# An Improved αvβ6-Receptor-Expressing Suspension Cell Line for Foot-and-Mouth Disease Vaccine Production

**DOI:** 10.3390/v14030621

**Published:** 2022-03-16

**Authors:** Yongjie Harvey, Ben Jackson, Brigid Veronica Carr, Kay Childs, Katy Moffat, Graham Freimanis, Chandana Tennakoon, Nicholas Juleff, Julian Seago

**Affiliations:** 1The Pirbright Institute, Ash Road, Woking GU24 0NF, UK; kitty.harvey@pirbright.ac.uk (Y.H.); ben.jackson@pirbright.ac.uk (B.J.); veronica.carr@pirbright.ac.uk (B.V.C.); kay.childs@pirbright.ac.uk (K.C.); katy.moffat@pirbright.ac.uk (K.M.); graham.freimanis@pirbright.ac.uk (G.F.); chandana.tennakoon@pirbright.ac.uk (C.T.); 2Bill & Melinda Gates Foundation, 500 5th Ave. N, Seattle, WA 98109, USA; nick.juleff@gatesfoundation.org

**Keywords:** foot-and-mouth disease, FMDV, vaccine, suspension BHK-21

## Abstract

Foot-and-mouth disease (FMD) is endemic in large parts of sub-Saharan Africa, Asia and South America, where outbreaks in cloven-hooved livestock threaten food security and have severe economic impacts. Vaccination in endemic regions remains the most effective control strategy. Current FMD vaccines are produced from chemically inactivated foot-and-mouth disease virus (FMDV) grown in suspension cultures of baby hamster kidney 21 cells (BHK-21). Strain diversity means vaccines produced from one subtype may not fully protect against circulating disparate subtypes, necessitating the development of new vaccine strains that “antigenically match”. However, some viruses have proven difficult to adapt to cell culture, slowing the manufacturing process, reducing vaccine yield and limiting the availability of effective vaccines, as well as potentiating the selection of undesired antigenic changes. To circumvent the need to cell culture adapt FMDV, we have used a systematic approach to develop recombinant suspension BHK-21 that stably express the key FMDV receptor integrin αvβ6. We show that αvβ6 expression is retained at consistently high levels as a mixed cell population and as a clonal cell line. Following exposure to field strains of FMDV, these recombinant BHK-21 facilitated higher virus yields compared to both parental and control BHK-21, whilst demonstrating comparable growth kinetics. The presented data supports the application of these recombinant αvβ6-expressing BHK-21 in future FMD vaccine production.

## 1. Introduction

Foot-and-mouth disease (FMD) is a highly contagious viral infection of cloven-hooved livestock that is endemic in many regions of the world, including most of sub-Saharan Africa and large parts of Asia and South America. FMD threatens global food security, and its presence in endemic regions has an estimated annual impact of US$6.5–21 billion [1]. The causative agent, FMD virus (FMDV), exists as seven distinct serotypes, O, A, C, Asia 1 and Southern African Territories (SAT) 1, 2 and 3, with each serotype containing numerous and evolving subtypes generated through recombination and error-prone replication [2]. Vaccination remains the most effective tactic for controlling FMD in endemic regions, with about 2.5 billion doses used annually, and current vaccines are made from inactivated preparations of whole FMDV propagated in suspension cultures of baby hamster kidney-21 cells (BHK-21) [1]. Vaccination against one serotype does not provide efficacious cross protection against another and often not to disparate strains within the same serotype either. In the case of the SAT serotypes, the strains that are included within a topotype have nucleotide differences of up to 20% in the VP1-coding sequence [3]. In the absence of a vaccine that “antigenically-matches” the outbreak strain, a new vaccine seedstock must be rapidly developed. This involves selection of FMDV field strains for use as candidate vaccine seedstocks followed by cell culture adaptation to enable efficient BHK-21 infection and cost-effective large scale vaccine production. However, adaptation of some FMDV field strains to cell culture can be challenging, leading to the production of sub-optimal virus yields or the alternative use of inadequately “matched” vaccine strains. Non-cell culture-adapted FMDV use integrin receptors of the αv subgroup to gain entry into their target host cells, specifically αvβ1, αvβ3, αvβ6 and αvβ8 [4,5,6,7]. Cell culture adaptation of such non-adapted strains leads to the selection of variants within the viral quasispecies that can utilize heparan sulphate (HS) proteoglycans (HSPG) as secondary receptors for infection [8,9]. To date, there have been numerous approaches to cell culture adaptions of FMDV strains, including sequential passaging within and between cell lines, and the use of reverse genetics to insert targeted mutations, informed from sequencing such passaged virus, into the external capsid-coding regions of the genome [10,11,12,13]. These approaches have had good success but are time-consuming and the targeted capsid mutations required for cell culture adaptation may be multiple and serotype or strain specific [10]. In addition, these cell culture adaptation methods can affect virus phenotype, including capsid stability and antigenicity [14,15,16,17,18,19].

To circumvent the need to cell culture adapt candidate FMD vaccine seedstock from different serotypes, and to improve the yields of existing vaccine strains that have not fully adapted to BHK-21 suspension culture, we decided to initiate the development of suspension BHK-21 permissive to infection by both cell culture-adapted as well as non-cell culture-adapted FMDV. To do this, we used a systematic approach that involved (1) the suspension adaptation of four adherent BHK-21 cell lines, (2) the subsequent selection of two suspension BHK-21 cell lines based on their capacity to support the production of FMDV and (3) the use of these selected cell types to generate recombinant suspension BHK-21 that over-express the integrin αvβ6. Importantly, we show that the over-expression of αvβ6 in these recombinant suspension BHK-21 is stable over multiple cell passages and does facilitate improved virus yields following infection with non-cell culture-adapted FMDV. Additionally, we show that the repertoire of integrin receptors expressed by adherent BHK-21 is variable and cell line specific, and that suspension adaptation influences integrin expression.

## 2. Materials and Methods

### 2.1. Cell Lines

Adherent BHK-21 cell types (BHK 21 (Clone-13), 85011433; BHK 21 Strain 31, 93120840; BHK 21 Strain 35, 93120841; BHK 21 Strain 38, 93120842) and suspension BHK-21 cell types (BHK21C13-2P, 84111301; BHK/AC9, 85040103) were obtained from HPA Culture Collections supplied by Merck (Merck Life Science UK Ltd., Dorset, UK). The biological source of these cell lines is Syrian Hamster Kidney. All cell lines were grown in Glasgow’s Modified Eagle Medium MEM (GMEM) + 2 mM Glutamine (Thermo Fisher Scientific, Paisley, UK) containing 10% adult bovine serum (Selborne Biological Services Ltd., Alton, UK), 0.5% HyPep^®^ 4601 Protein Hydrolysate from wheat gluten (Merck) or EX-CELL^®^ CD Hydrolysate Fusion (Merck). Adherent cell lines were suspension adapted in Erlenmeyer flasks on a Celltron shaker (INFORS HT, Reigate, UK) in the presence of 0.1% Pluronic^®^ F68 (Merck, P5556), 5% CO_2_ at 37 °C. Cells were initially seeded at 5 × 10^5^ cells/mL and considered suspension adapted once cell density had reached 1 × 10^6^ cells/mL with 90% viability by day 3 post-seeding for 3 passages. Seeding density was then reduced to 2 × 10^5^ cells/mL. Cells were regularly tested for Mycoplasma. Antibiotics (penicillin (100 SI units/mL and streptomycin (100 μg/mL), Merck) were used in infection experiments but were not used in the preparation of cell seedstocks.

### 2.2. Construction of αvβ6 Integrin Expression Plasmids and Transfection of Cells

The open reading frames (ORFs) encoding hamster αν (hαν) and β6 (hβ6) subunits were ordered from GeneArt (Thermo Fisher Scientific, Paisley, UK). The hαν ORF was PCR-cloned into the pcDNA3.1 plasmid vector to generate pcDNA3.1/hα. The hβ6 ORF was PCR-cloned into the pLJM1 lentiviral vector (Nova Lifetech Inc., Hong Kong, China) to generate pLJM1/hβ6 and then co-transfected with packaging plasmids (pLP1, pLP2 and VSV-G) into HEK-293T cells to generate lentiviruses. To generate hανβ6-expressing cells, BHK-21 were first transfected with pcDNA3.1/hαν, selected for 3 weeks in the presence of G418 (Merck) and then infected with hβ6-expressing lentivirus. Lentiviruses were added to BHK-21 in the presence of 2 µg/mL Polybrene (Merck, H9268) and centrifuged at 1000× *g* relative centrifugal force (rcf) for 30 minutes (min). At 72 h post infection (hpi), cells were treated with 3 µg/mL puromycin (Thermo Fisher) for a further 72 h to select for transduced cells. Forty-eight hours after removal of selection, surviving cells were positively sorted for αvβ6 expression.

### 2.3. Characterisation of Cell-Surface Expression of Integrin Heterodimers by Flow Cytometry

Adherent cells were removed from the flasks using a cell dissociation solution (Merck). For each analysis with either primary mAb or corresponding isotype-matched control antibody, 1 × 10^6^ cells were incubated in blocking buffer (PBS containing magnesium and calcium, normal goat serum 10% (*w*/*v*)) for 15 min prior to the addition of the respective antibody at 10µg/mL for 15 min at room temperature. Primary mAbs used: anti-integrin β1 (clone 9EG7, 553715, IgG2a; BD Pharmingen, San Diego, CA, USA), anti αvβ3 (clone LM609, MAB1976, IgG1; Merck), anti-αvβ5 (clone P1F6, MAB1961, IgG1; Merck), anti-α5β1 (clone SAM-1, ab6131, IgG2b; Abcam, Cambridge, UK), anti-αvβ6 (clone 10D5, ab77906, IgG2a; Abcam), anti-integrin β8 (clone ADWA 16, IgG1; gift from Amha Atakilit at the University of California, San Francisco) and anti-heparan sulphate (clone JM403, 370730-1, IgM; AMSBIO, Abingdon, UK). Primary isotype-matched control antibodies: TRT1 (IgG1) and TRT3 (IgG2a) [20], AV29 (IgG2b) [21] and anti-mouse IgM isotype control (Agilent Technologies Inc., Santa Clara, CA, USA). After washing twice with complete PBS, the cells were then incubated with the corresponding isotype FITC-conjugated goat anti-mouse IgG antibody (1/200; SouthernBiotec, Birmingham, AL, USA) for 15 min at room temperature in the dark. After the final wash, the cells were resuspended in 1% paraformaldehyde (PFA) before acquisition using an LSRFortessa™ flow cytometer (BD Biosciences, San Jose, CA, USA). Data were collected using DIVA 8 acquisition software and analyzed in FCSexpress (version 7, De Novo software). A minimum of 30,000 events were collected for each sample. The gating strategy used to identify αvβ6 cells was as follows: samples were gated on cells (SSC-A vs. FSC-A) and singlets (SSC-A vs. SSC-H) and αvβ6 expression was identified as cells positive for FITC. Positive integrin surface marker expression was determined against appropriate isotype control staining in overlaid histograms.

For sorting, cells were passed through a 70 μm cell strainer (BD Biosciences). Alphavβ6 positive cells were identified and collected, using DIVA 8 acquisition software and a FACS Aria UIII cell sorter (BD Biosciences). The gating strategy used to identify αvβ6 cells was as follows: samples were gated on cells (SSC-A vs. FSC-A) and singlets (SSC-A vs. SSC-H) and αvβ6 expression was identified as cells positive for FITC. BHK21C13-2P cells line were used to determine the αvβ6 threshold. Positive integrin surface marker expression was determined against appropriate isotype control staining in overlaid histograms.

### 2.4. Immunofluorescence Microscopy

Cells prepared on cover glasses were fixed with 4% PFA (Santa Cruz Biotechnology, Dallas, TX, USA) for 20 min and blocked in 10% normal goat serum (G9023; Merck) in PBSa (lacking MgCl_2_ and CaCl_2_). A primary mAb recognizing αvβ6 (clone 10D5, ab77906; Abcam Ltd.) and a secondary goat anti-mouse IgG antibody conjugated with Alexa Fluor 488 (Thermo Fisher) were used diluted in blocking buffer at dilutions of 1 in 200 and 1 in 500, respectively. Nuclei were stained with DAPI (4′,6-diamidino-2-phenylindole; Sigma).

### 2.5. FMDV Strains

Non-cell culture and cell culture-adapted variants of the wild type FMDV strains O/ETH/29/2008, A/ETH/9/2008, SAT1/KEN/80/2010 and SAT2/ETH/65/2009 have recently been described [22]. Similarly, the recombinant non-cell culture and cell culture-adapted FMDV strains, termed Nano-FMDV and Nano-FMDV-HS+, respectively, have been described [23].

### 2.6. Plaque Assays

Ten-fold virus dilutions were used to infect triplicate wells of confluent ZZ-R 127 cells pre-seeded in 6-well tissue culture plates. Following adsorption at 37 °C for 1 h, the inoculum was removed and 2 mL of indubiose (MP Biomedicals, Santa Ana, CA, USA) overlay was added. Following incubation (48 h, 37 °C, 5%CO_2_), cells were fixed by the addition of 10% Tetrachloroauric acid (Merck) for 30 min. Indubiose plugs were then removed and cells were stained with methyl blue solution (PBS, 10% formaldehyde, 10% of 1% methyl blue in ethanol) prior to determining plaque forming units/mL (PFU/mL).

### 2.7. Virus Neutralisation Test

Existing Day 21 monovalent vaccinate sera from groups of five cattle vaccinated against O/ETH/29/2008, A/ETH/9/2008, SAT1/KEN/80/2010 or SAT2/ETH/65/2009 were used for the homologous virus neutralization test (VNT) [22]. Homologous VNT were conducted according to the protocol recommended by the World Organisation for Animal Health (Office International des Epizooties (OIE)) [24] and as previously described [22], aside from the use of SP38-Mix cells in place of adherent BHK-21 cells. Neutralizing antibody titres, calculated by the Spearmann–Karber method, were expressed as the last dilution of serum that neutralizes 50% of the virus [25].

### 2.8. Luciferase Assays

NanoLuc luciferase in cell culture supernatants obtained from cell cultures at the indicated time points post infection was quantified using the Nano-Glo™ Luciferase Assay System (Promega, Chilworth, UK) in triplicate according to the manufacturer’s instructions and using a Promega GloMax microplate reader.

### 2.9. Statistical Analyses

Statistical analyses were carried out using Excel. Data are presented as mean ± standard deviation (SD). An unpaired Student’s T-test with unequal variance was used to compare two groups. In all figures, the statistical significance between the indicated samples and control is designated as * *p* < 0.05.

## 3. Results

### 3.1. Adaptation of BHK-21 to Suspension Culture and Effect on FMDV Receptor Expression

To develop an improved suspension BHK-21 cell line permissive to infection by both cell culture adapted as well as non-cell culture-adapted FMD vaccine strains, four BHK-21 adherent cell lines were obtained from the European Collection of Authenticated Cell Cultures (ECACC) and propagated with agitation until they were adapted to suspension (SP) culture; the four cell lines were BHK 21 Strain 31 (ECACC 93120840), BHK 21 Strain 35 (ECACC 93120841), BHK 21 Strain 38 (ECACC 93120842) and BHK 21 Clone 13 (ECACC 85011433). Although initial differences were observed, all four cell lines (termed SP Strain 31, SP Strain 35, SP Strain 38 and SP Clone 13, respectively) exhibited a doubling time of 24 h by 5 consecutive passes and were thus considered suspension adapted.

Next, to determine the FMDV receptor repertoire expressed on the surface of each of the four adherent BHK-21 cell lines, and to investigate the effects of adaptation to suspension culture, flow cytometry analyses were performed using a panel of antibodies recognizing integrins αvβ3, αvβ6 and αvβ8. A suitable antibody recognizing integrin αvβ1 was not available; therefore, analyses were performed using an anti-β1 antibody. Additional analyses were performed to determine the expression of heparan sulphate on the surface of each cell type, as well as the integrins α5β1 and αvβ5 (Table 1). Isotype antibodies served as controls in the analyses and were used as a comparison to determine the respective percentage of cells expressing each receptor. Interestingly, differences in the expression of the integrin panel were observed between the adherent cell lines as well as following their adaptation to suspension culture. Apart from αvβ8, which was reduced following suspension adaptation of Clone 13, Strain 35 and Strain 38, the percentage of cells expressing each integrin increased following suspension culture adaptation, albeit marginally in most cases. Comparing the suspension cell lines that had been generated, SP Strain 38 exhibited the highest percentage of cells expressing the FMDV receptors αvβ3 (18.48%) and αvβ6 (3.70%), whilst SP Strain 31 exhibited the highest for αvβ8 (6.92%). Both SP Strains 31 and 35 exhibited comparatively higher levels of β1 (21.42% and 21.04%, respectively). To facilitate comparison with an existing BHK-21 SP cell line, BHK21C13-2P (ECACC 84111301) cells were analyzed using the same panel of antibodies. Similar to the SP Clone 13 suspension cells developed in this study, BHK21C13-2P was derived from the adherent BHK-21 Clone 13 cell line (ECACC 85011433); comparison of these two suspension lines showed BHK21C13-2P exhibited lower levels of β1 (4.78% vs. 16.18%), higher αvβ3 (11.57% vs. 4.03%) and comparable but negligible levels of αvβ6 (5.09% vs. 3.41%) and αvβ8 (3.88% vs. 5.47%). All the adherent and SP cell lines expressed integrin αvβ5 (44.11–75.86%) and the heparan sulphate (32.96–58.78%) receptors. Of note, in comparison to the other adherent cell lines, Strain 38 and Clone 13 exhibited higher levels of αvβ3 and αvβ8, respectively.

To assess the ability of the new SP cell lines to support FMDV production, single infections (MOI 0.1) were carried out using cell culture-adapted FMDV belonging to the O, A, SAT1 and SAT2 serotypes, and at 24 h post infection (hpi), virus yields were determined by plaque assay. For comparison, BHK21C13-2P cells were infected alongside and their respective virus yields determined. Surprisingly, each new SP cell line produced significantly higher yields of the four-cell culture-adapted FMDV strains in comparison to BHK21C13-2P (Figure 1). However, the four respective virus yields produced by an individual SP cell line differed by up to 4 logs. SP Strain 31, 35 and 38 produced their highest virus yields for SAT1, SP Clone 13 and BHK21C13-2P for SAT2, whilst yields of O FMDV were the lowest for all cell types. Based on their comparatively high yields of all four viruses, SP Strain 35 and SP Strain 38 were selected for further development of an improved cell line for FMD vaccine production.

### 3.2. Generation of Recombinant Suspension BHK-21 Expressing αvβ6 Integrin

It was hypothesized that increased integrin expression would facilitate cell susceptibility to infection by non-cell culture-adapted FMDV. SP Strain 35 and SP Strain 38 were therefore used to generate recombinant cell types expressing the key FMDV receptor, αvβ6 integrin. We reasoned that a comparatively higher level of αvβ6 expression may be achieved by using open reading frames (ORFs) encoding the αv (XP_012976915) and β6 (XP_021082235.1) subunits of the golden (Syrian) hamster integrin, as the β6 subunit would likely pair with both exogenous αv, as well as endogenous αv subunits expressed as a component of other receptors such as αvβ5. Stable populations of SP Strain 35 and SP Strain 38 cells expressing hamster αvβ6 were subsequently generated by transduction with a lentivirus expressing the β6 subunit, followed by transfection with a plasmid expressing the αv subunit. These recombinant cell populations were amplified and positively enriched for cells expressing αvβ6 by fluorescence-activated cell sorting (FACS); the enriched cell populations of SP Strain 35 and SP Strain 38 that expressed αvβ6 were termed SP35-Mix and SP38-Mix, respectively. Flow cytometry analyses using a monoclonal antibody (mAb) recognizing αvβ6 subsequently revealed respective expression levels of 50.07% and 87.45% for the SP35-Mix and SP38-Mix populations (Figure 2a). In comparison, the parental SP Strain 35 and SP Strain 38 cell lines (negative controls) exhibited respective αvβ6 expression levels of 10.43% and 1.63% (Figure 2a). In two further analyses, the parental SP Strain 35 cell line exhibited slightly reduced αvβ6 expression levels of 7.4% and 8.1%, respectively (Appendix A). Immunofluorescence confocal microscopy using the same mAb confirmed a higher level of αvβ6 expression on the surface of SP38-Mix compared to SP35-Mix (Figure 2b). No staining was observed for parental SP Strain 38, but in agreement with the flow cytometry analyses, parental SP Strain 35 exhibited a low level of staining (Figure 2b). As the SP38-Mix cell population exhibited a comparatively higher level of αvβ6 expression, it was selected for further characterization and the production of clonal cells.

FMD vaccine production requires cell consistency over multiple passages; therefore, stable expression of αvβ6 on the cell surface of SP38-Mix was assessed by flow cytometry (Figure 2c) over thirty consecutive passages (2–3 passages per week). As expected, in comparison to passage 1 (75.84%), some variation in αvβ6 expression was observed during the experimental period, but importantly, expression of the integrin was retained throughout the experiment and the percentage of cells expressing αvβ6 at passage 30 (87.11%) was higher in comparison to that at passage 1.

To investigate the diversity of αvβ6 expression in the cell population, and to generate a collection of monoclonal cell lines, endpoint dilution of SP38-Mix was carried out. The subsequent expansion of single cells yielded seventy putative monoclonal suspension BHK-21 cell lines, which were divided into four sets and analyzed on separate occasions along with the controls by flow cytometry to determine their respective cell surface expression of αvβ6 and their clonal status (Appendix A). Three of the cell lines were deemed to consist of two cell populations (non-monoclonal), as judged by the presence of two flow cytometry peaks, and two cell lines expressed negligible levels of αvβ6 (2.38% and 4.63%). Of the remaining sixty-five monoclonal cell lines, all exhibited αvβ6 expression levels > 20%, 62 > 50% and 32 > 80%. Three monoclonal cell lines with low (26.37%), medium (51.40%) or high (98.49%) αvβ6 expression were selected to investigate comparative virus yields; these lines were termed SP38-Low, SP38-Med and SP38-High, respectively. As the SP38-High cell line exhibited particularly high αvβ6 expression, the receptor was quantitatively confirmed by flow cytometry over twenty consecutive passages (2–3 passages per week); an expression level of 92.9–97.2% was observed over five analyses (Appendix A).

### 3.3. Evaluation of Suspension BHK-21 Expressing αvβ6 Integrin to Support Non-Cell Culture Adapted FMDV

The susceptibility of the three αvβ6-expressing monoclonal cell lines (SP38-Low, SP38-Med and SP38-High) to infection by non-cell culture-adapted FMDV was next investigated using a recombinant virus termed Nano-FMDV that cannot infect adherent BHK-21 cells [23]. Additionally, a recombinant virus termed Nano-FMDV-HS+, which can infect adherent BHK-21 cells via the heparan sulphate receptor, was used as a control [23]. Both these recombinant FMDVs express the nanoluciferase reporter protein, thus facilitating quantification of viral replication [26]. To evaluate cell susceptibility, infections were carried out at a low MOI (0.001) and cell supernatants were assayed for nanoluciferase activity at 24, 48, 64 and 72 hpi as a readout for virus replication (Figure 3a,b); SP Strain 38, SP38-Mix and BHK21C13-2P cells were infected alongside for comparison. Following infection with Nano-FMDV-HS+, the analyses of collected supernatants from the non-recombinant parental (SP Strain 38) and four recombinant (SP38-Mix, SP38-Low, SP38-Med, SP38-High) cell types showed a comparable increase in nanoluciferase activity over the duration of the experiment; as expected, these results confirmed that all the Strain 38-derived cell types had facilitated equivalent uptake of the cell culture-adapted virus regardless of their respective level of αvβ6 expression. Interestingly, only a low level of nanoluciferase activity, which did not increase over the duration of the experiment, was observed for BHK21C13-2P, suggesting a low level of infection had occurred (Figure 3a). Infection with Nano-FMDV yielded similar results, with the exception that no nanoluciferase activity was detected following infection of parental SP Strain 38, thus confirming a lack of virus replication in the absence of recombinant αvβ6 expression (Figure 3b). To corroborate these results with the production of infectious virus, we used the cell culture supernatants collected at 48 and 72 hpi to perform plaque assays on the ZZ-R 127 goat cell line that expresses the principal FMDV receptor, integrin αvβ6 (Figure 3c,d). In agreement with the Nanoluciferase-results, all the αvβ6-expressing cell lines supported virus production following infection with either Nano FMDV or Nano FMDV HS+, whilst parental Strain 38 only supported the production of Nano FMDV HS+. As predicted by the comparably low levels of luciferase activity that were observed following infection of BHK21C13-2P, virus progeny was only detected (15 PFU/mL) at 72 hpi with Nano FMDV HS+.

To further investigate the ability of the αvβ6-expressing cell types to support the production of non-cell culture adapted FMDV, additional infection experiments were performed using a low MOI (0.005) and FMDV strains belonging to the O, A, SAT1 and SAT2 serotypes; these FMDV had been passaged (<3 passages) in primary bovine thyroid (BTY) cells following collection from the field and were considered not to be cell-culture adapted. Parental SP Strain 35 and 38, as well as BHK21C13-2P, were infected alongside the SP38-Mix, SP38-Low, SP38-Med and SP38-High cells for comparison, and 24 and 48 hpi cell culture supernatants were quantified by plaque assay titration on ZZ-R 127 cells (Figure 3e). In comparison to SP Strain 38, the αvβ6-expressing cell types produced higher virus yields of O, A, SAT1 and SAT2 FMDV strains at 24 and 48 hpi; statistical analyses using unequal variance deemed all but 3 of these yields to be significantly higher. In agreement with our previous results, the BHK21C13-2P cell line was unable to support the production of non-cell culture-adapted FMDV. Interestingly, at 48 hpi, the non-recombinant parental lines (SP Strain 35 and 38) produced higher yields of SAT1 compared to O, A and SAT2 FMDV, a phenomenon that was also observed following their infection with cell culture-adapted virus (Figure 1). With regard to virus yields produced at 48 hpi by the αvβ6-expressing cell types, SP38-Low produced the highest yield of O FMDV and SP38-Mix the highest of SAT2, whilst both these cell types shared the highest yields of A FMDV. Comparable SAT1 yields were produced at 48 hpi by all four tested αvβ6-expressing cell types.

### 3.4. Comparative Growth Analysis of Suspension BHK-21 Expressing αvβ6 Integrin

Next, the suspension growth of each αvβ6-expressing cell type (SP38-Mix, SP38-Low, SP38-Med or SP38-High) was compared against the non-recombinant parental line (SP Strain 38), as well as SP Strain 35 and BHK21C13-2P. To do this, small scale cultures were seeded (1 × 10^5^ cells/mL) in rotating Erlenmeyer flasks and the cell density was determined every 24 h for 4 days. Final cell densities (live cells) ranged from 0.49–2.32 × 10^6^ cells/mL, with SP38-Low exhibiting the lowest and SP38-Mix the highest cell density, respectively (Figure 4).

To compare growth, rotating Erlenmeyer flasks were individually seeded (1 × 10^5^ cells/mL) with an αvβ6-expressing cell type (SP38-Mix, SP38-Low, SP38-Med or SP38-High), the non-recombinant parental line (SP Strain 38), SP Strain 35 or BHK21C13-2P and cell density was determined every 24 h over 4 days.

### 3.5. Additional Applications of SP38-Mix: Virus Neutralisation Test and Detection of Cytopathic Effect

To investigate if the SP38-Mix cells have potential applications other than vaccine production, they were used to perform homologous virus neutralization tests (VNT) and as a diagnostic tool to detect cytopathic effect (CPE). Notably, non-cell culture-adapted (<3 passages in BTY) viruses were used for both experiments. Homologous VNT were conducted using four FMDV strains (O/ETH/29/2008, A/ETH/9/2008, SAT1/KEN/80/2010 and SAT2/ETH) and corresponding monovalent vaccinate sera (5 sera per FMDV strain) [22]. Interestingly, these VNT generated mean neutralizing antibody titres that were comparable to recently reported VNT results obtained using the same panel of vaccinate sera, but adherent BHK-21 (BHK-21 Clone 13) not expressing αvβ6 and cell culture-adapted stocks of the same four FMDV strains (Appendix A) [22].

To facilitate CPE diagnosis, the SP38-Mix cells were first incubated in stationary flasks for 5 consecutive passages to develop an adherent phenotype; these cells were termed Ad38-Mix and exhibited a flattened morphology comparable to other adherent BHK-21 lines (Appendix A). CPE assays were then performed by infecting monolayers (MOI 0.01) of Ad38-Mix with SAT2/ETH/65/2009 in tissue culture tubes and subsequent incubation on a rotary platform. FMDV-sensitive ZZ-R 127 goat cells were infected under the same conditions as a control. Comparable levels of CPE were observed in ZZ-R 127 goat and Ad38-Mix cells at 30 hpi (Appendix A).

## 4. Discussion

Industrial FMD vaccine production utilizes suspension BHK cells but requires the cell culture adaptation of vaccine FMDV strains to enable the use of the heparan sulphate (HS) proteoglycan for cell entry. However, the cell culture adaptation of FMDV can be problematic and time consuming and may result in the production of sub-optimal virus yields and affect antigenicity. Unlike previous approaches that have focused on FMDV adaptation through either reciprocal passaging or reverse genetics, we used a systematic approach to develop new cell lines with increased susceptibility to infection by non-cell culture-adapted virus. Initially, four different BHK-21 cell types were adapted to suspension culture and screened by assessing virus yield following infection with cell culture-adapted FMDV strains. Based on these findings, two suspension BHK-21 cell types were selected for use in the generation of recombinant BHK-21 populations that over-express hamster αvβ6 integrin, and from these, the SP38-Mix cell type was selected for further characterization.

To facilitate the over-expression of αvβ6 integrin in the cell lines, hamster ORFs encoding the αv and β6 subunits were used. Although we selected αvβ6 for overexpression in suspension BHK-21, FMDV can initiate infection via the recognition of one of at least four different cell-surface integrin molecules, αvβ1, αvβ3, αvβ6 or αvβ8, by a highly conserved Arg-Gly-Asp (RGD) amino acid sequence motif located in the G-H loop of the VP1 capsid protein. Within the animal host, the αvβ6 interaction is believed to be the most relevant [27].

In this study, the expression of FMDV receptors by each cell type was quantified by flow cytometry before and after suspension adaptation. Following adaptation, a considerable increase in β1 expression, with marginal increases in the expression of α5β1, αvβ3 and αvβ6, were observed by all cell types, although the expression of α5β1, and importantly αvβ6, remained negligible. The expression of αvβ8 was more varied, with 3 of the 4 cell types displaying a reduction. Interestingly, the high level of αvβ3 expression exhibited by adherent Strain 38 was retained through this process. In addition to integrin receptors, the Jumonji C-domain containing protein 6 (JMJD6) has been proposed to facilitate cell entry by FMDV in an integrin- and heparan sulphate-independent manner [28]. We did not detect JMJD6 when whole cell lysates of BHK-21 were analyzed by Western blot (data not shown).

Differences in integrin usage between serotypes have been observed in vitro. Duque and colleagues investigated the usage of αvβ1/β3/β5/β6 by serotype O and A FMDV [29]. While all the viruses could infect cells expressing these integrins, they exhibited different efficiencies of integrin utilization. All the type A viruses used αvβ3 and αvβ6 with relatively high efficiency, while only one virus utilized αvβ1 with moderate efficiency. In contrast, both type O viruses utilized αvβ6 and αvβ1 with higher efficiency than αvβ3. Only low levels of viral replication were detected in αvβ5-expressing cells infected with either serotype. In agreement, Berinstein reported that antibodies to α5β1 or to the integrin αvβ5 had no effect on either binding or plaque formation [30]. Under specific conditions, purified preparations of the human α5β1 have been shown to bind FMDV; however, the relevance of α5β1 for infection in vivo remains to be determined [31]. In this study, no obvious correlation between the expression of a specific integrin and virus yield was observed prior to αvβ6 overexpression; this is likely due to the expression of heparan sulphate by all cell types and the use of cell culture-adapted viruses. However, it is feasible that the comparatively higher expression of αvβ3 by SP38 and of β1 by SP31 and SP35 may have contributed to their respective yields of A, SAT1 and SAT2 (Figure 1). Of note, the infection of non-recombinant suspension cells with cell culture-adapted O FMDV produced lower virus yields compared to infection with cell culture-adapted A, SAT1 or SAT2. Importantly, in subsequent infections that utilized αvβ6-expressing BHK-21 cell types and non-cell culture-adapted virus, yields of O FMDV were more comparable to those of A/SAT1/SAT2; indeed, titres of >10^7^ PFU/mL were consistently obtained, highlighting the role of αvβ6 as a key integrin for this serotype.

Other researchers have implemented the over-expression of integrin to produce cells that are more susceptible to FMDV infection, but these studies have not used suspension cells or BHK-21 [32,33,34]. La Rocco analyzed the susceptibility of six cell types (parental LFBK, LFBK-αvβ6, LK, IBRS-2, MVPK and BHK) to animal-derived FMDVs (two A serotypes, four O, two Asia1, two C, one SAT1, one SAT2 and one SAT3) [33]. Although virus replication was supported in each line, differences in virus titres were observed both between cell types and between different strains, and the porcine LFBK cell line overexpressing bovine αvβ6 (LFBK-αvβ6) was shown to produce the highest virus titre for each strain.

To confirm that the recombinant SP38 cell types expressing αvβ6 are more susceptible to infection by non-cell culture adapted FMDV in comparison to parental SP Strain 38, we utilized field viruses that had been minimally passaged in BTY, and Nano-FMDV that expresses the quantifiable nanoluciferase marker protein. We have previously shown that Nano-FMDV can only infect BHK-21 following targeted amino acid mutation (VP3 56 His to Arg) that facilitates heparan sulphate-mediated uptake [23,35].

The SP38-Mix cell population was used to develop monoclonal lines that exhibited a range of αvβ6 expression levels; this phenomenon was likely due to the integration of the ORFs encoding the integrin subunits into varied sites of the host’s genome. Comparative analyses of the virus yields produced by SP38-Mix and three monoclonal lines (SP38-Low/-Med/-High) exhibiting differential expression levels of αvβ6 showed that SP38-Mix was able to support the production of four different FMDV serotypes to a comparable level or higher than that of the monoclonal lines. In comparison to the same monoclonal lines, SP38-Mix also exhibited better growth. Surprisingly, although the SP38 cell population is comprised of cells that express αvβ6 at different levels, it maintained a mean level of αvβ6 expression of 81.34 ± 5% over 30 passages (7 flow cytometric analyses, Figure 2).

Interestingly, the SP38-Mix cells were successfully used to conduct homologous VNTs without the need for cell culture adaptation of challenge viruses. Indeed, we have recently shown that three (O/ETH/29/2008, A/ETH/9/2008 and SAT1/KEN/80/10) of the four challenge viruses used for these VNTs exhibited amino acid changes in their external capsid proteins during adaptation to adherent BHK-21 Clone 13 cells [22]. Although the VNT assay is based on the use of adherent cells, SP38-Mix cells were able to form adherent cell sheets in the absence of shaking over the 3-day incubation period of the protocol. In agreement, Ad38-Mix was produced from SP38-Mix simply by stationary incubation. Although Ad38-Mix appeared equally sensitive to FMDV infection, as judged by CPE, Ad38-Mix cells are smaller than ZZ-R 127 cells and this may impede the detection of small areas of CPE. Additionally, cultures of Ad38-Mix grew noticeably faster than ZZ-R 127 cells in our hands, and consequently reduced the pH of media more rapidly. As FMDV is sensitive to mildly acidic pH values, this may hinder the detection of CPE over extended periods of cell incubation.

In summary, we have developed recombinant suspension BHK-21 cell populations and cell lines with increased susceptibility to infection by non-cell culture adapted virus and shown these have promising applications to the development of new FMD vaccine seedstocks and associated production of conventional FMD vaccines.

## Figures and Tables

**Figure 1 viruses-14-00621-f001:**
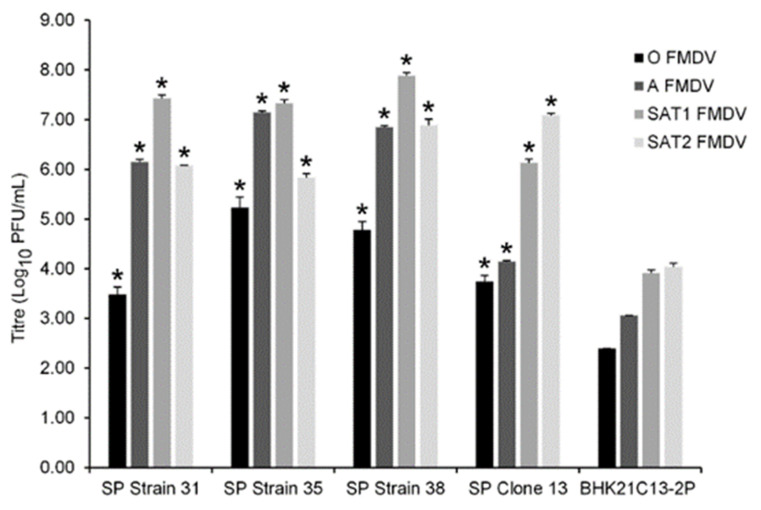
Comparison of FMDV yields produced by suspension BHK-21. Following adaptation to suspension culture, SP Strain 31, SP Strain 35, SP Strain 38 and SP Clone 13 were individually infected (MOI 0.1) with cell culture adapted O/ETH/29/2008, A/ETH/9/2008, SAT1/KEN/80/2010 or SAT2/ETH/65/2009 FMDV strains. BHK21C13-2P were infected alongside and served as a comparative suspension cell line. At 24 hpi, progeny virus in the respective cell culture supernatant was quantified by plaque assay titration. These data are representative of two separate experiments performed in triplicate. Standard deviation bars are shown. Student’s T-test with unequal variance was used to compare respective virus yields between each cell type and BHK21C13-2P; * *p* < 0.05.

**Figure 2 viruses-14-00621-f002:**
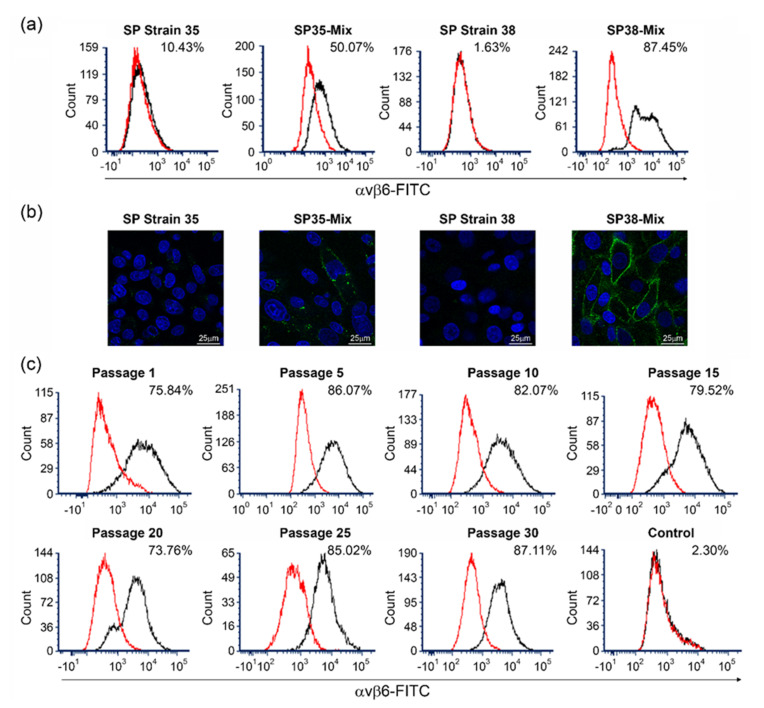
Analysis of αvβ6 expression in parental and recombinant cell populations. Parental SP Strain 35 and SP Strain 38, and recombinant SP35-Mix and SP38-Mix cell populations were assessed for αvβ6 expression by (**a**) flow cytometry analysis and (**b**) immunofluorescence confocal microscopy (αvβ6 in green) using mAb clone 10D5. Nuclei are stained blue with DAPI. (**c**) To assess the stability of αvβ6 expression, SP38-Mix was consecutively passaged 30 times and flow cytometry analysis was performed every fifth passage using mAb clone 10D5. (**a**,**c**) The percentage (%) of positive cells expressing αvβ6 in each cell population is indicated. Cells were stained with anti-αvβ6 (clone 10D5, black) or an isotype control (red).

**Figure 3 viruses-14-00621-f003:**
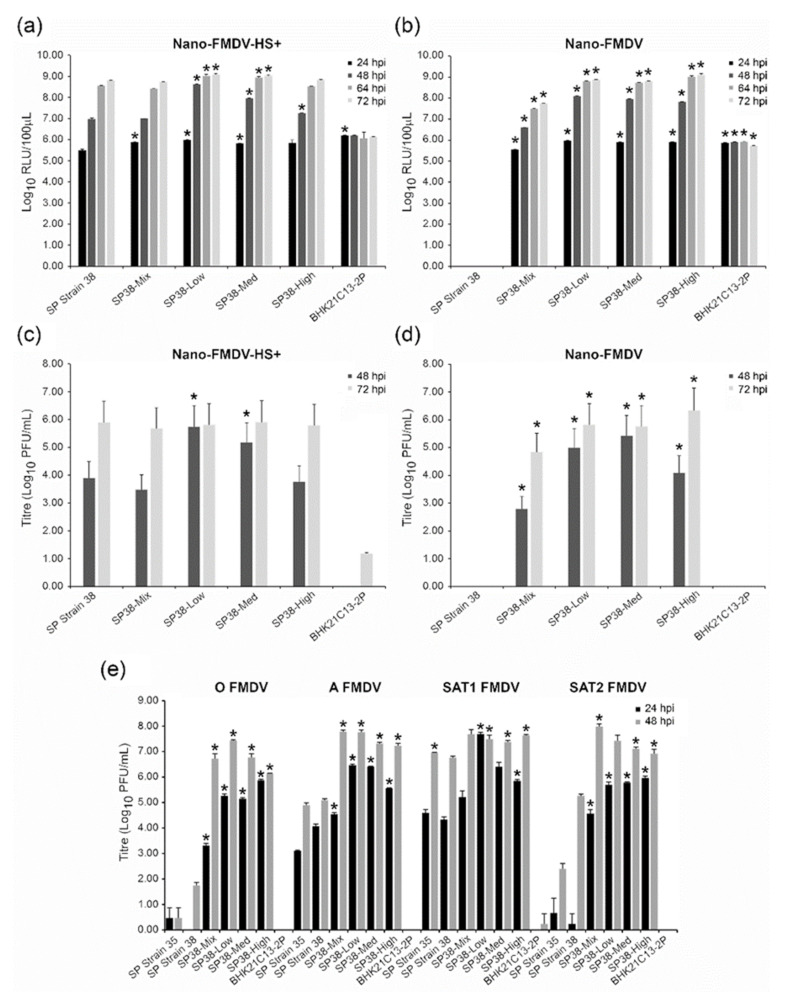
Comparison of virus yields following infection with non-cell culture-adapted FMDV. (**a**,**b**) Four αvβ6-expressing suspension cell types (SP38-Mix, SP38-Low, SP38-Med and SP38-High) were infected (MOI 0.001) with either cell culture-adapted Nano-FMDV-HS+ (**a**) or non-cell culture-adapted Nano-FMDV (b) and at 24, 48, 64 and 72 hpi, the respective cell culture supernatant was assayed for Nanoluciferase activity to assess virus replication. Parental SP Strain 38 and BHK21C13-2P were infected alongside for comparison. (**c**,**d**). To quantify the respective virus yields produced from these infections, the cell culture supernatants collected in (**a**,**b**) at 48 and 72 hpi were analyzed by plaque assay ((**c**,**d**) respectively). (**e**). The same four αvβ6-expressing suspension cell types, in addition to parental SP Strain 38, BHK21C13-2P SP and SP Strain 35 were infected (MOI 0.005) with either O/ETH/29/2008, A/ETH/9/2008, SAT1/KEN/80/2010 or SAT2/ETH/65/2009. At 24 and 48 hpi, cell culture supernatants were collected and analyzed by plaque assay to determine the respective virus yields. These data are representative of two separate experiments performed in triplicate. Student’s T-test with unequal variance was used to compare respective virus yields between each cell type and SP Strain 38; * *p* < 0.05.

**Figure 4 viruses-14-00621-f004:**
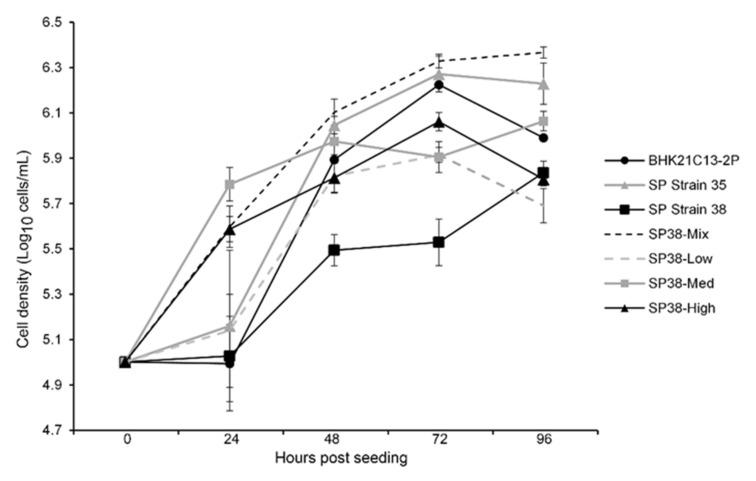
Comparative growth analysis of parental and recombinant suspension BHK-21.

**Table 1 viruses-14-00621-t001:** Flow cytometry analyses of integrin and heparan sulphate expression on the cell surface of BHK-21 cell lines. Four adherent cell lines (BHK-21 Strain 31, 35 and 38, and BHK 21 Clone 13), and the respective suspension cell lines generated from them (SP Strain 31, 35, and 38, and SP Clone 13) were analyzed by flow cytometry using a panel of antibodies recognizing α5β1, β1, αvβ3, αvβ5, αvβ6, αvβ8 and heparan sulphate. BHK21C13-2P was analyzed alongside and served as a comparative suspension cell line. Matched isotype antibodies served as negative controls and the expression of each receptor is presented as the percentage of positive cells.

BHK-21 Cell Line	Percent (%) of Cells Positive in Comparison to Isotype Control						
	α5β1	β1	αvβ3	αvβ5	αvβ6	αvβ8	Heparan Sulphate
Adherent Clone 13	1.32	4.31	0.41	57.69	0.03	16.39	58.74
Adherent Strain 31	0.04	2.76	4.83	44.11	0.02	5.29	51.67
Adherent Strain 35	0.25	4.55	2.49	55.82	0.07	6.57	34.53
Adherent Strain38	0.19	3.27	15.67	47.26	0.07	1.38	45.56
Suspension Clone 13	2.15	16.18	4.03	60.91	3.41	5.47	40.88
Suspension Strain 31	1.30	21.42	6.99	75.86	1.61	6.92	48.25
Suspension Strain 35	2.75	21.04	4.70	72.24	1.95	0.00	32.96
Suspension Strain 38	1.03	12.08	18.48	50.20	3.70	0.00	46.88
Suspension BHK21C13-2P	4.66	4.78	11.57	65.92	5.09	3.88	58.78

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
