# Peer review of "An Improved αvβ6-Receptor-Expressing Suspension Cell Line for Foot-and-Mouth Disease Vaccine Production"

_viruses, 2022, doi:10.3390/v14030621_

Round 1

Reviewer 1 Report

Commercial currently available FMD vaccines are preparations of inactivated FMD virus in formulation with adjuvants. Usually, vaccine manufacturers use suspension BHK cells (hamster) for most virus strains, but some viruses grow better than others in these cells. Some viral strains grow well on them and some need to be adapted by repeated passage, usually a few times, on BHK cells. This adaptation poses the risk of selection of adapted mutant strains that might not induce the same coverage as the original strain, due to antigenic changes resulting from adaptation. Over the years, in addition to perform good quality control, people have circumvented some of these problems, by adapting less the seed strains, or making vaccines with genetically engineered viruses containing not only the necessary mutations to adapt them to grow in hamster cells without significantly affecting their antigenicity, but also by including other desired features that would allow for DIVA (differentiation of infected from vaccinated animals). More recently people have shown that several commercially available vaccines could protect against other strains of viruses within the same serotype by just increasing the mass of the vaccine dose. Given the current stage of economic development, this has become a more common practice, and in general people use FMD vaccines at higher doses than before also including mixes of strains to provide broader and stronger coverage. Regardless, any adjustment that could potentially reduce costs may be welcomed for the target community.  

In this paper the authors derived suspension BHK cells with increased susceptibility to FMDV strains that had or had not been previously adapted to cell culture (such as field strains).  Following a similar strategy previously reported for other cells, the authors overexpressed integrins, the main natural receptor for the virus, in suspension BHK cells, and achieved similarly to the earlier published papers, an increase of virus yield as well as increased susceptibility to multiple strains of all FMDV serotypes.

The paper is clear, the authors demonstrate their hypotheses, and derive cells that might be of high interest, mostly to the community interested in manufacturing FMD vaccines.

Specific technical minor comments

It would be nice to compare FMDV capsid sequence data after passage in these cells as compared to currently used suspension and adherent BHK-21 cells.

Presumably, increased susceptibility should correlate with highest expression of integrins, therefore in Fig 3, one would have expected to see highest titers in the viruses grown in the SP high expressers. I am not sure I see any difference in susceptibility depending on the levels of integrin expression for each cell type. Besides, the mix seem to yield lower viral titers than the low expression clones. What do the authors think about these observations?

Although the cells seemed to have been best characterized for vaccine production, the authors talked and did experiments to show the potential of these cells in other assays such as detection of CPE during diagnostics or evaluation of virus neutralization tests in serum samples for potency or cross reactivity. Both these applications will require the adaptation from suspension to adherent cell phenotypes. Do the authors expect any change in susceptibility to field viruses?

What about  evaluation of cell susceptibility to viruses present in probang samples?

What about cross neutralization assays with non homologous strains?

Reviewer 2 Report

The manuscript by Harvey et al describes the generation and characterization of suspension BHK21 cell lines expressing avb6 integrin that can potentially amplify field FMD viruses without needing extensive adaptation of the virus to the cells. The study is very relevant and could potentially facilitate rapid production of homologous FMD vaccines for more efficient control of outbreaks. The manuscript is well written.

Please check the designation of the cells and keep it consistent. You have SP strain 38 at the beginning, then just strain 38 later and SP38 in some parts of the text.

Reviewer 3 Report

Review of paper 1586634

An improved αvβ6-receptor-expressing suspension cell line for foot-and-mouth disease vaccine production

Harvey et al

The elegant study used a systematic approach to develop suspension BHK cells that can potentially be used for FMD vaccine production. The paper is interesting and well written.

I have a few editorial suggestions and recommendations for the discussion.

Abstract line 11 and introduction line 29-31, FMD is not endemic in South America; most countries have controlled the disease very successfully and at present it is probably only Venezuela that is endemic.  Please change to reflect this.

Line 21: please mention that the hamster integrin receptor was expressed

Line 33: please mention that serotype C is most likely extinct

Line 68-69: change to read: these selected cell types to generate recombinant suspension BHK-21 that over-express the HAMSTER integrin αvβ6

Lines 96-88: please provide the volume of the cell culture adaption steps.

Lines 128-129: define SSC-A, FSC-A and SSC-H

Lines 151-153: please provide a very brief description of Nano-FMDV and Nano-FMDV-HS+

Line 156: please provide a reference for the ZZ-R 127 cells

Line 165-170: change to read: Existing Day 21 monovalent vaccinate sera from groups of five cattle vaccinated against O/ETH/29/2008, A/ETH/9/2008, SAT1/KEN/80/2010 or SAT2/ETH/65/2009 were used for homologous virus neutralisation test (VNT) conducted according to the protocol recommended by the World Organisation for Animal  Health (Office International des Epizooties (OIE)) [24] as previously described [22], aside from the use of SP38-Mix cells in place of adherent BHK-21 cells.

Lines 339-340: ‘As predicted by the comparably low levels of luciferase activity that were observed following infection of BHK21C13 2P, virus progeny were only detected (15 PFU/ml) at 72 hpi with Nano FMDV HS+’.  I could not detect this value in Fig 3d, is it because the level is too low to show on the chart?  Please mention this.

Figure 4: How many repeats were done of the comparative growth analysis?  Please provide.

The non-adapted viruses used in this study had been passed on BTY cells <3 passages.  Simply as a point of interest, did the authors consider testing their new cells to support virus isolation directly from clinical material?  I fully realise the aim of the study was not to develop a cell line for primary diagnosis, but it could be a useful tool for countries that cannot support primary cell culture production (although the apparent difficulty to observe CPE and the pH changes may be deterrents).  I would also appreciate a comment on the ease of use of suspension cell culture vs adherent cells for ease of use in other laboratories.  It is apparent from the results that the SP cells can be grown as adherent cells, but how stable will they be as adherent cells over a large number of passes?  A comment on these points in the discussion would be good.

The volume for the suspension cultures was not provided and should be.  I also ask that the authors provide a statement on upscaling of this technology.  Have you done any studies to investigate the stability of the integrin expression when reaching volumes related to vaccine production?  Even if the work has not been done, a comment on this needs to be provided.

Please also comment on the cell growth in relation to potential vaccine production, i.e. final concentration of cells and time to reach concentrations of cells suitable for production (lines 379-382; figure 4).

The La Rocco study used bovine integrins to increase cell susceptibility, whilst this study used the hamster integrins.  It would be interesting to get your views on whether one could expect differences between the integrins from cloven hoofed, susceptible animals vs a ‘laboratory model’ animal.